# Novel Treatment Targets Based on Insights in the Etiology of Depression: Role of IL-6 Trans-Signaling and Stress-Induced Elevation of Glutamate and ATP

**DOI:** 10.3390/ph12030113

**Published:** 2019-07-29

**Authors:** Hans O. Kalkman

**Affiliations:** Retired pharmacologist, Gänsbühlgartenweg 7, CH4132 Muttenz, Switzerland; hans.kalkman@bluewin.ch; Tel.: +41-61-362-0110

**Keywords:** major depressive disorder, suicide, inflammasome, ketamine, vagus nerve

## Abstract

Inflammation and psychological stress are risk factors for major depression and suicide. Both increase central glutamate levels and activate the hypothalamic-pituitary-adrenal axis and the sympathetic nervous system. Both factors also affect the function of the chloride transporters, Na-K-Cl-cotransporter-1 (NKCC1) and K-Cl-cotransporter-2 (KCC2), and provoke interleukin-6 (IL-6) trans-signaling. This leads to measurable increases in circulating corticosteroids, catecholamines, anxiety, somatic and psychological symptoms, and a decline in cognitive functions. Recognition of the sequence of pathological events allows the prediction of novel targets for therapeutic intervention. Amongst others, these include blockade of the big-K potassium channel, blockade of the P2X4 channel, TYK2-kinase inhibition, noradrenaline α2B-receptor antagonism, nicotinic α7-receptor stimulation, and the Sgp130Fc antibody. A better understanding of downstream processes evoked by inflammation and stress also allows suggestions for tentatively better biomarkers (e.g., SERPINA3N, MARCKS, or ^13^C-tryptophan metabolism).

## 1. Introduction

Major depressive disorder (MDD) is worldwide a leading cause of years lived with disability [1,2]. Unfortunately, drug development in psychiatry has stagnated during the last decades [3,4]. However, continuing research has led to the recognition of the importance of the immune system in psychiatric disorders [5,6], including depression [7,8]. Based on insights provided by ongoing academic research efforts, it is possible to reinterpret the wealth of information on the pathophysiological processes in major depressive disorder. This better understanding enables proposals for novel pharmacological treatment targets. The current review starts off with a reinterpretation of existing disease knowledge. This is subsequently used to list hitherto underexplored pharmacological treatment options. Finally, it provides proposals for novel biomarkers for endophenotypic depression symptoms, as well as suicide risk.

## 2. Stress as a Risk Factor for Depression and Suicide

Stress, including early life stress, is an important risk factor for major depression [9,10,11,12,13,14,15]. Early life stress (childhood neglect, physical or sexual abuse, or early parental loss) not only constitutes a major risk factor for depression, but also significantly increases suicide risk [16,17]. Stressors like family or romantic conflicts and legal or disciplinary problems can trigger suicidal behaviors [17] and manic episodes [18,19]. Stress-induced depression is characterized by deep mental and physical fatigue, disturbed non-restorative sleep, irritability, emotionality, concentration problems and memory disturbances [9]. Psychological stress can be investigated under laboratory conditions by a tool called the Trier social stress test (TSST) [20]. Application of the TSST frequently evokes marked increases in the activity of the sympathetic nervous system and the hypothalamic-pituitary-adrenal (HPA) axis (seen as increases in circulating levels of catecholamines, adrenocorticotrophic hormone (ACTH), and cortisol, and tachycardia) [20,21]. Interestingly, the circulating levels of interleukin-6 (IL-6) are also increased, as is the activity of the pro-inflammatory nuclear-factor kappa-B (NFκB)-pathway in monocytes [21,22,23,24]. Notably, TSST-induced pro-inflammatory activity is particularly strong in individuals with a history of childhood maltreatment [22]. Both the corticosteroid “stress”-hormones and the sympathetic nervous system represent important effector systems. Chronic social stress in laboratory animals and in humans has been shown to stimulate gene transcription of pro-inflammatory proteins in monocytes via an activation of β-adrenoceptors [25,26] and α1B receptors [27]. Repeated social defeat in mice has been observed to increase the number of monocytes trafficking to the brain, while plasma levels of IL-6 and the weight of the spleen have been seen to increase [28]. Blockade of β-adrenoceptors during the defeat period has been shown to prevent these changes [28]. Notably, the observation that stress leads to activation of the immune system has been conceptualized as a preparation for subsequent wounding and possible infection [14,15]. This interpretation is supported by the observation that defeat-stress in mice improves bacterial clearance by phagocytes [29]. As in peripheral monocytes, in the central nervous system stress can result in activation of microglial cells [30,31,32] (reviewed by [5,33,34]). Stress-induced microglia activation is microscopically visible in terms of an alteration of the microglia phenotype [35,36].

### 2.1. Stress Modifies KCC2 and NKCC1 Activity

The direction of anion-flux through the gamma-aminobutyric acid-A (GABA_A_) channel depends on the electrochemical gradient for the chloride ion and is determined by the expression and activity of two trans-membrane chloride-transporters, K-Cl-cotransporter-2 (KCC2) and Na-K-Cl-cotransporter-1 (NKCC1) (reviewed by [37]). KCC2 extrudes chloride, while NKCC1 performs chloride-influx. A reduction in the expression of KCC2 or the inhibition of the activity impairs chloride-extrusion, which leads to higher intracellular chloride levels [37]. Under such conditions, a GABA-induced opening of the GABA_A_-channel causes a chloride efflux and thus a depolarization. 

Corticotrophin-releasing factor (CRF)-producing cells within the paraventricular nucleus (PVN) are activated by glutamate-neurons and noradrenergic neurons [9,38,39] but under inhibitory control by GABA-neurons [38,40,41]. An altered direction of the chloride flux through GABA-channels thus has consequences for the activity of the HPA axis.

A variety of experiments in rodents show that stress affects the function of chloride transporters. For instance, acute restraint stress in rats has been shown to activate α1-adrenoceptors in the PVN [40], which suppressed the activity of the KCC2. Consequently, restraint stress was observed to lead to a depolarization shift, rendering GABA-agonists activating rather than inhibitory [40]. In mice, acute restraint stress has been seen to reduce the cell-surface expression of KCC2 [41]. The modified chloride-gradient has been observed to cause an increase in excitatory GABA transmission. Interestingly, under this particular condition, GABA_A_-mimetic neurosteroids were found to raise the firing frequency of CRF neurons and caused significant increases in circulating corticosterone-levels [41]. Moreover, the anxiolytic activity of neurosteroids has been seen to be altered to an anxiogenic effect [41]. In a model of chronic stress, Gao and colleagues [42] found that repeated exposure to unpredictable mild stress increased the protein levels of NKCC1 in CRF neurons of rat PVN. The NKCC1 levels remained elevated for at least 10 days [42]. Moreover, the authors confirmed that acute restraint stress decreased KCC2 levels in the PVN; however, this effect was short-lasting and returned to baseline within 5 days post stress. These data indicate that acute stress impairs GABA control over CRF neurons via a decrease in KCC2 function, whereas chronic stress-induced impairment is mediated via an increase in NKCC1 expression. The depolarizing activity of GABA is not restricted to the PVN and similar effects have been noticed in the spinal cord and are thought to play a role in neuropathic pain and hyperalgesia [43,44]. The effect of stress on chloride transporters and consequently on GABA-function may provide a mechanistic explanation for “the GABAergic deficit hypothesis of major depressive disorder” [14,45]. A dysfunctional GABA-inhibition would lead to a diminished suppression of glutamate neurons and to an increased activity of the sympathetic nervous system and HPA axis. The reader may notice that this process represents a vicious circle. A further vicious circle may be formed when high levels of cortisol activate the mineralo-corticoid receptor (MR) because MR-activation is known to increase the metabolic stability of NKCC1 [46]. 

### 2.2. Hyperactivation of the HPA Axis

A prolonged activation of the HPA axis may result in an exhaustion (atrophy) of cortisol production by the adrenal. Low circulating levels of cortisol are observed in a number of chronic stress disorders in humans. This includes atypical depression [47], posttraumatic stress disorder [48], and suicide attempts [49]. Since activation of the HPA axis inhibits immune-cell function [15,50,51], an exhausted cortisol production might lead to a hyper-immune state with increased levels of IL-6 [47,52]. 

### 2.3. Stress Induces Increased Central Glutamate Signaling

Immobilization stress in rats has been found to significantly increase extracellular glutamate levels in the hippocampus and frontal cortex [53,54]. A role of corticosterone in this response is indicated by the observation that adrenalectomy markedly attenuates the stress-induced increase in glutamate [53,54]. Additionally, chronic unpredictable mild stress has been observed to cause an increase in extracellular glutamate levels in these two brain areas, acting on N-methyl-D-aspartate (NMDA) receptors that couple to the NFκB signaling pathway [55]. Furthermore, foot-shock stress, as applied in the learned helplessness model, has been shown to cause a marked increase in neuronal glutamate release in the rat frontal cortex, which again is dependent on glucocorticoid receptor (GR) activation [56]. Finally, chronic mild stress has been found to increase the expression of NMDA-NR1 receptors on CRF neurons in the rat PVN, which in turn leads to an increased excitation of the HPA axis [39]. These data indicate that stress causes a corticosterone-driven increase in neuronal glutamate release [57]. Neuron-derived glutamate provokes adenosyl-triphosphate (ATP)-release by astrocytes, which represents a danger-signal for microglial cells [58]. Chronic psychosocial stress decreases astroglial plasticity in the rodent hippocampus and frontal cortex [59,60]. In another experiment, four sessions of restraint stress elevated corticosterone levels in mice in vivo. Blockade of corticosterone synthesis, blockade of the glucocorticoid receptor by RU486, or blockade of NMDA-receptors by MK801 prevented stress-induced microglia proliferation [50]. MK801 also prevented microglia-proliferation following exogenous corticosterone administration to non-stressed mice [50]. These results suggest that stress activates the GR receptor on neurons, which leads to increased extracellular levels of glutamate. This glutamate may subsequently activate NMDA receptors on astrocytes, and these then produce ATP (see Figure 1). Data by Ferrini and De Koninck indicate that ATP activates P2X4 receptors on microglia, and this provokes the release of brain-derived neurotrophic factor (BDNF) [44]. BDNF will activate tropomyosin receptor kinase-B (TrkB) receptors on neurons and reduce the activity of KCC2 (promoting GABA-induced depolarization) [44]. This summary indicates that stress in the CNS triggers a sequence of events that ultimately leads to an increase in neuronal network excitability [44]. 

ATP also stimulates microglia to assemble the components of the ‘NOD-, LRR- and pyrin-domain-containing protein 3′ (NLRP3) inflammasome [58]. The NLRP3 inflammasome mediates the cleavage of pro-interleukin-1β (pro-IL1β) to mature IL1β [34] and sensitizes microglia to generate a stronger pro-inflammatory response [61]. Frank et al. have exposed rats to inescapable tail shocks. They noted that the responsiveness of hippocampal microglia to lipopolysaccharide (LPS) was raised by the tail-shock procedure, but, notably, this sensitization was absent if the rats had been adrenalectomized or treated with the GR-antagonist RU486 prior to the tail shocks [61]. Chronic stress reduced the dendritic connectivity and size of the hippocampus and prefrontal cortex (PFC) [62,63,64], while inhibition of NMDA receptors or inhibition of glutamate-release blocked the effect of chronic stress on dendritic atrophy (reviewed by [57]). Stress-induced reduction in neuropil is likely to contribute to impairment in cognitive function [65,66].

### 2.4. Stress Increases Levels of Interleukin-6

In human subjects, physical and psychosocial stress can cause central and peripheral IL-6 release [15,67,68,69]. Daily life stressors evoked a particularly large increase in IL-6 in individuals who had suffered from childhood trauma [22,70]. In addition, rodent studies provide ample evidence for stress-induced IL-6 release. For instance, chronic unpredictable mild stress has been shown to elevate the levels of the IL-6 protein in the hippocampus and alter animal behavior [71]. Treatment with the NMDA-blocker ketamine has been found to provoke a rapid reduction in circulating IL-6 levels and normalized ‘depressed’ behavior [71]. Chronic mild stress, similar to other forms of stress such as acute foot shock or chronic intermittent cold exposure, has been seen to increase IL-6 levels in the hypothalamus of rats and activate the IL-6 – gp130 – STAT3 (signal transducer and activator-3) signaling pathway [72]. In apparent contradiction to these results is the observation that rats which are resilient to learned helplessness (a further rodent model of depression) have lower brain levels of IL-6 than their helpless counterparts [73]. The solution to this apparent conundrum is that IL-6 signals in two fundamentally different ways. When IL-6 acts via the membrane-bound IL-6 receptor (called “classical signaling”) it mainly causes neuron-protective effects [74,75,76]. However, when IL-6 acts in conjunction with the membrane-shed moiety of the IL-6 receptor (soluble IL6R, or sIL6R), a process called “trans-signaling”, it provokes mainly pathologic effects [75,77,78,79]. A detailed discussion about factors that influence the use of the two signaling modes is provided in Section 3.1 below.

## 3. Inflammation as a Risk Factor for Depression and Suicide

Whereas the previous sections dealt with the effect of stress on depression-related parameters, the next sections deal with inflammation. It may be noted that central and peripheral inflammation cause effects that resemble those that are due to stress. Miller and Raison [80] have formulated the “pathogen defense hypothesis of depression”, which posits that risk alleles for depression are the ones that inhibit growth of pathogens by a pro-inflammatory activity, and therefore are conserved in the human genome (Table 1).

Depression is comorbid with numerous disorders involving inflammation (Table 1; for reviews see [6,14,15,95,96]). Conversely, inflammation markers such as IL-6 are often elevated in depression. For instance, depressed patients with metabolic syndrome [97] or patients suffering from atypical depression (e.g., hyperphagia, weight gain, hypersomnia) display high plasma levels of IL-6 [98]. Meta-analyses of cytokine levels show that plasma IL-6 levels [99,100,101,102], tumor necrosis factor-α (TNFα) [99,100], and circulating levels of C-reactive protein (CRP) [101,102] are elevated in patients with MDD. Moreover, IL-6 has been identified as an important susceptibility gene for major depression [103]. Serum IL-6 and sIL6R have also been found to be higher after delivery, especially in women with a history of depression [104]. Clinical studies measuring sIL6R in unipolar depression are still sparse, but two meta-analyses in bipolar depression have reported that circulating levels of IL-6 and sIL6R were higher in patients than in healthy controls [105,106]. Clearly elevated levels of sIL6R in serum and cerebrospinal fluid (CSF) have been observed in patients with neurological inflammatory diseases such as multiple sclerosis [107]. In a systematic literature search of studies concerning cytokine levels in patients with suicidal ideation, suicide attempts, or suicide completion, elevated IL-6 in CSF, blood, and postmortem brain tissue was found in 8 out of 14 studies [108]. In suicide attempters, plasma IL-6 levels and CSF IL-6 levels did not correlate, and, interestingly, IL-6 levels were higher in plasma than in CSF ([109,110] but see also [111]). High plasma levels of IL-6 were associated with increased suicidal ideation [112] and suicide attempts [113], and they were independent of depression severity [112,113,114]. Moreover, IL-6 levels in plasma are also associated with suicide endophenotypic behaviors, such as increased extraversion, impulsivity, and violent attempts [109,115,116]. Measurements of sIL6R levels in individuals showing suicidal behavior have, apparently, not yet been performed [117]. Apart from these clinical studies, rodent depression models such as chronic unpredictable mild stress, learned helplessness, maternal separation, forced swim test/tail suspension test, prenatal stress, and olfactory bulbectomy are also associated with significant increases in IL1β, TNFα and IL-6 in the brain and blood [68,118].

### 3.1. IL-6 Trans-Signaling in Depression

Although leukocytes, fibroblasts, adipocytes, keratinocytes, and endothelial cells all secrete IL-6 [119], about 30% of circulating IL-6 is derived from adipose tissue [13]. Homeostatic production of IL-6 leads to plasma levels of 1–10 pg/mL, but during infection, inflammation, or cancer, these levels are elevated to the lower ng/mL range. Signal transduction by IL-6 involves the formation of a hexamer built from IL-6, the membrane-bound IL-6-receptor, and the gp130 protein [74,120,121]. This form of signaling is occasionally called “cis”-signaling but is more often referred to as “classical”-signaling. Expression of the IL-6-receptor (IL6R) is confined to neutrophils, monocytes, CD4 T-cells (but not CD8 T-cells), memory T-cells, and hepatic and osteoblast cell lines [122,123]. In the brain, IL6R is strongly expressed by microglia, but very weakly (or not at all) by astrocytes, oligodendrocytes, and neurons [77,124,125]. Two metallo-proteinases, ‘a desintegrin and metalloproteinase-17′ (ADAM17) and ADAM10, are able to cleave the extracellular part (the ‘ectodomain’) of the IL-6-receptor [74,122,126,127]. The resulting moiety (soluble IL6R, or ‘sIL6R’) still binds IL-6, and the IL-6/sIL6R complex can activate cells that express gp130, notably with no need for membrane-bound IL6R [122,127]. This is called IL-6 “trans-signaling” ([122]; for a review see [76]). ADAM10 and ADAM17 activity is induced by phorbol-esters, by the cytokines IL1β and TNFα, and by apoptotic pathways (e.g., DNA-damage, UV radiation, and Fas ligation) [127]. Additionally, CRP provokes an increase in sIL6R production [123,128]. Hence, in principle, sIL6R can be produced by the sympathetic nervous system (via the α1 receptor-Gq-protein-kinase-C (PKC)-diacylglycerol (DAG) signaling pathway (mimicked by phorbolesters)) and during inflammation (when levels of CRP, TNFα, or IL1β are elevated). Hepatocytes, neutrophils, and CD4^+^ T-cells represent the major sources of circulating sIL6R [122,127,128]. In human serum sIL6R is always present at relatively high concentrations of 25–75 ng/mL, and these levels are 2–3 fold increased during inflammation [119]. Soluble-IL6R circulates at elevated levels in various diseases [128], including major depressive disorder [129] and bipolar disorder [105,106]. The ratio of IL-6 to sIL6R/IL-6 determines to which degree trans-signaling will occur [119,123]. Importantly, circulating sIL6R may cross the blood brain barrier and cause IL-6 trans-signaling in the CNS [130].

### 3.2. IL-6 Trans-Signaling in the Brain

Mice with a genetic overexpression of IL-6 by astrocytes have been found to respond to restraint stress with an exaggerated rise in plasma corticosterone [131]. This is consistent with data in humans that IL-6 activates the HPA axis [132]. Since this is a neuronal response to IL-6, it is likely that it involves trans-signaling via sIL6R. Inflammation in the CNS leads to production of reactive oxygen species, whereas oxidative stress is known to decrease in cell-surface expression due to a rapid decline in KCC2 tyrosine-phosphorylation [133]. In addition, in sensory nerves it has been shown that IL-6 signaling also alters phosphorylation of NKCC1, which in this case led to higher cell-surface expression and higher intracellular chloride levels [134]. It is conceivable that IL-6 might provoke similar effects in central neurons. As discussed in earlier sections, both mechanisms, a decline in KCC2 and an increase in NKCC1, would contribute a depolarizing activity of GABA_A_. IL-6 trans-signaling therefore increases the synaptic excitation/inhibition ratio [135,136]. It is likely that IL-6 will not only enhance the activity of the HPA axis but also the activity of the sympathetic nervous system (c.f. [137]). Peripheral and central inflammation in animals indeed increases the activity of the sympathetic nervous system [138,139,140].

Under physiological conditions, astrocytes assume numerous supportive functions, including structural support, neurovascular coupling, regulation of extracellular K^+^, uptake of neurotransmitters, and metabolic support of neurons [141]. Astrocytes are essential for regulation of glucose uptake and lactate release, uptake of glutamate and release of glutamine (required for glutamate as well as GABA neurotransmission), and uptake of glutathione precursors and the release of glutathione [141,142]. During brain inflammation, however, microglia is polarized to the activated M1-phenotype, and this, in turn, stimulates astrocytes to attain what is called a “neurotoxic reactive”, or “A1”-phenotype ([143]; see Figure 2). A1 astrocytes secrete an unknown factor that is highly toxic to a subset of neurons and to mature oligodendrocytes [143,144]. The sIL6R/IL-6 complex is a conceivable candidate for this toxic factor, but this has not been tested yet. Since corticosteroids inhibit the pro-inflammatory phenotype of microglial cells [145], it is likely that during brain inflammation corticosteroids are neuroprotective, which is in contrast to the stress-induced pathology described in Section 2.3. This makes the glucocorticoid receptor both a part of the problem and a part of the solution [146].

The data summarized in this section show that IL-6 in the pro-inflammatory trans-signaling mode provokes much of the downstream effects that are also observed after exposure to stress. It is evident that both stress and inflammation lead to activation of the sympathetic nervous system, the HPA axis, and an enhanced GABA-depolarization (presumably leading to anxiety, psychopathology, and cognitive decline), as well as somatic ‘sickness’ symptoms (see Figure 3). Although causing a similar spectrum of symptoms, it is likely that the two inputs, stress or inflammation, will differ in the intensity of their biological outputs [47,147]. It will be interesting to read future scientific literature to see if it fits the idea that melancholic and atypical depression are the respective outputs of the stress and the inflammation arms.

## 4. Potential for Therapeutic Intervention

### 4.1. Intervening in the Sequence of Events Provoked by Stress

Stress has a profoundly negative effect on the viability of neurons (Figure 1). The mechanism has been investigated in detail in in vitro experiments [135,148,149]. Corticosterone-induced neuron death is associated with a decrease in the activity of protein kinase-B (PKB or Akt) and an increase in glycogen-synthase kinase-3 (GSK3)-activity [135,148,149]. Factors that increase PKB-activity like leptin [135], insulin-like growth factor-1, [149], or inhibition of GSK3 with lithium, inhibit neuronal loss [148]. Zhang et al. ([150]) describe that dexamethason-induced apoptosis of neonatal hippocampal neurons involves the assembly of the NLRP1 inflammasome via an increased K-efflux through ‘big-K’ (BK) potassium channels. The effect of dexamethason was ascribed to an immediate effect on the electrophysiology of BK-channels but also to an increase in mRNA and protein levels of BK following a chronic (28 days) treatment of mice in vivo [150]. The GR-antagonist RU486 and the big-K inhibitor iberiotoxin blocked dexamethason-induced apoptosis [150]. This short summary shows that negative effects of stress can be diminished by multiple mechanisms, including GSK3-inhibition and blockade of the big-K potassium channel. These represent potential drug-development targets for treatment of depression.

The next logical target for interruption of the sequence of events shown in Figure 1 is NMDA blockade. The antidepressant activity of the NMDA-channel blocker ketamine is now well established but there are still doubts as to whether the antidepressant activity is mediated by NMDA-inhibition [57,151,152]. Consistent with data in Figure 1, the antidepressant effect of ketamine involves a rapid increase in the expression of BDNF and subsequent TrkB receptor activation [57,152,153]. However, another NMDA-channel blocking drug, memantine, has been observed to fail to produce antidepressant activity in humans (reviewed by [151]). Whilst BDNF expression and TrkB-activation is induced also by effective treatments like electroconvulsive shock therapy or tricyclic antidepressants [153], memantine has been shown to fail to induce BDNF transcription [151]. This result indicates that the degree of NMDA blockade by memantine might be too small. Another argument that casts doubt on the NMDA-mechanism of action is the observation that a metabolite of ketamine, 2R,6R-hydroxy-norketamine, lacks affinity for the ketamine-binding site but has still been found to generate a rapid antidepressant-like effect in animal studies [152]. This latter argument is not as strong, since it may well be that the metabolite simply acts via a different pharmacological activity (e.g., via activation of opiate μ-receptors; see [152]). BDNF decreases the function of the chloride extruder KCC2, thus promoting a polarizing activity of GABA [43,44,154,155]. It has been argued that a high intracellular chloride concentration is beneficial for the formation of novel dendrites and synapses [134,156].

The last evident target from the cascade pictured in Figure 1 is the P2X4 channel. Three subunits of P2X are required to form a functional channel and each has to be stimulated by ATP to open the channel [157]. Due to its effects on the NLRP3 inflammasome-formation in microglia and macrophages, the P2X4-related (P2X7) channel is a well-known target for depression, and several selective P2X7-inhibitors are currently under development [157]. The P2X4 channel is mostly known for its role in the development of neuropathic pain, and some lead compounds for drug development have been described [157,158,159]. Development hurdles are selectivity, bioavailability, brain penetration, and poor water solubility, as well as species differences in pharmacology [157,158,159]. Importantly, the blockade of P2X4 prevents BDNF-release and this ultimately results in a reduction in mRNA levels of KCC2 [159].

### 4.2. Intervening in the Sequence of Events Provoked by Inflammation

The sequence of events shown in Figure 2 is by no means as thoroughly validated as the one discussed in Section 4.1. Nevertheless, concerning depression, there can be no doubt that IL-6 trans-signaling has important pathophysiological consequences (see Section 3.1 and Section 3.2). In the IL-6-receptor family, the receptors for leukemia inhibitory factor, ciliary neurotrophic factor, and others associate with the Janus-kinases JAK1 and JAK2 [160], but gp130 exclusively associates with the tyrosine kinase-2 (TYK2) [161]. Since the IL-6 receptor is non-signaling, this implies that IL-6 in both classic and trans-signaling modes requires TYK2 for downstream signaling. Therefore, one way to block pathological IL-6 trans-signaling would be via TYK2 kinase-inhibition. Large pharmaceutical companies apparently have realized that selective TYK2 inhibition (vis-à-vis other kinases, in particular also to JAK1 and JAK2) may represent a worthwhile development target [161,162,163]. However, an obvious disadvantage of this approach is that it will also inhibit the desirable IL-6 classic signaling. Antibodies directed against IL-6 or IL6R suffer from the same disadvantage. Importantly though, this is not the case for a recombinant derivative of the soluble gp130 protein, “sgp130Fc” [76]. Sgp130Fc is sgp130 bound to the Fc portion of IgG, and acts as a specific inhibitor of IL-6 trans-signaling [127]. During sepsis, sgp130Fc has been shown to inhibited sIL6R signaling while the anti-inflammatory classic signaling remained unaffected and regenerative proliferation was retained [76,127]. Sgp130Fc (Olamkicept^®^ Conaris/Ferring) is in phase II clinical trials. Sgp130fc has shown efficacy in numerous preclinical inflammation models, including CNS inflammation [76], but its effect in depression models remains to be tested. Apart from the need for a parenteral route of administration, a further potential issue could be an insufficient blood-brain passage for those cases where the depressogenic inflammation is within the central nervous system.

### 4.3. Common Pathways Activated by Stress and Inflammation

A further useful approach to inhibit inflammation is by stimulation of the vagus nerve [164] or mimicking the effects of the vagus by nicotinic α7-agonists [165,166,167]. Electrical stimulation of the efferent vagus nerves in rats prevented LPS-induced endotoxic shock [168] and reduced secretion of IL-6 and TNFα [169]. It has been suggested that exercise, controlled breathing, relaxation therapies and fish-oil increase the activity of the vagus nerve and decrease production of TNFα and IL-6 [164]. Electrical stimulation of the vagus nerve has been shown to improve major depression in treatment-resistant patients [164,170,171,172]. There is furthermore an extensive preclinical literature that stimulation of the nicotinic α7-receptor with agonists like acetylcholine, choline, carbachol, nicotine, or the relatively selective agonist GTS-21 causes a reduction in inflammation-induced cytokine-release from human and rodent microglia cells and macrophages [165,167,168,172,173,174,175,176]. Although shown thus far during brain maturation only, activation of nicotinic α7-receptors has the propensity to modulate chloride transporter levels [177,178]. This makes nicotinic α7 receptor activation an exciting target for drug development in depression, but up to now no development compound has been tested for depression and as of the year 2019, no compound has reached the market.

A more global overview of the down-stream consequences of stress and inflammation is provided in Figure 3. Cognitive function is worsened by acute uncontrollable stress exposure and involves an increase in noradrenaline/α1-adrenoceptor/PKC activation in the prefrontal cortex (reviewed by [62]). This has led to the use of the α1-adrenoceptor antagonist prazosin for treatment of stress-related disorders [62]. Centrally acting α2-adrenoceptor agonists like clonidine or guanfacine are alternatives for prazosin [62]. These compounds reduce the activity of the sympathetic nervous system [179] while stimulating the activity of the parasympathetic nervous system [180]. Unfortunately, their profound hypotensive effect [179] is a disadvantage for use in psychiatric indications. There are three subtypes of α2 receptors in human genome and of these it is the α2A subtype that seems responsible for cardiovascular activity [181]. Notably, the α2B receptor has received considerable attention, because a mutation in the α2B receptor gene was associated with higher performance in a cognitive task for emotional stimuli and a stronger emotional memory [182,183,184]. The mutant form of the receptor (a deletion of three amino acids) is relatively resistant to receptor-desensitization [185] and gives rise to an elevated sympathetic outflow [186]. These findings suggest that a selective α2B-receptor antagonist could be useful to dampen excessive responding to emotional stimuli, in particular in homozygous carriers of the mutant form of the α2B-receptor.

There are a number of ways in which high intracellular chloride levels can be reduced. The most obvious is the inhibition of the NKCC1 transporter. This can be done with registered drugs like the diuretic bumetanide (review by [187]) or, interestingly, by oxytocin-spray [187]. A further possibility is inhibition of kinases like lysine deficient kinase (WNK) and SPS1-related proline/alanine-rich serine-threonine kinase (SPAK), as they regulate the cell surface-persistence of NKCC1 and KCC2 [188]. However, seemingly, the discovery and development of such kinase-inhibitors is not far advanced.

## 5. Biomarkers

### 5.1. Biomarker for IL-6 Trans-Signaling

In contrast to the psychological symptoms of depression (mood symptoms, anxiety, irritability, and cognitive alterations), the inflammation-induced vegetative symptoms (flue-like symptoms, fatigue, or anorexia) respond poorly to treatment with antidepressants [15,189,190]. This implies that a full remission of depression is achieved only when the driving force behind the vegetative symptoms is eliminated. IL-6 in its trans-signaling mode is the prime suspect for this symptom cluster (see Figure 3). Indeed, successful antidepressant treatment is associated with a reduction in plasma IL-6 levels [100,129,191,192,193,194,195]. However, since IL-6 levels do not necessarily reflect the degree of trans-signaling, one would need a marker for trans-signaling. Cellular internalization of the complex IL-6/sIL6R is slower than that of IL-6 bound to membrane-localized IL6R. Therefore, IL-6 trans-signaling leads to stronger and longer lasting intracellular signaling than IL-6 classic signaling [76]. SERine proteinase inhibitor-A3N (SERPINA3N) is an example of a gene that is regulated by trans-signaling but not via classic signaling [77], and probably there are more examples to be found. These genes could serve as biomarkers for the sickness syndrome-related symptoms of depression and suicide risk [123].

### 5.2. Biomarkers for Stress-Induced Effects

As a biomarker for the stress pathway to depression, one could propose salivary α-amylase [196,197]. Whether salivary α-amylase (sAA) is also a useful marker for suicidality is questionable, since relatives of suicide completers unexpectedly have been found to display a blunted stress-induced sAA response [198]. Stress-induced activation of the sympathetic nervous system can lead to activation of PKC, and this may promote sIL6R shedding and cognitive decline (see above). One of the substrates of PKC is the protein myristoylated alanine-rich C-kinase substrate (MARCKS) [199,200]. Interestingly, the expression level of MARCKS has been shown to be consistently increased in circulating leukocytes in different cohorts of suicidal bipolar depression-patients [201]. As a read-out of peripheral activity of the sympathetic nervous system, it might therefore be worthwhile to study phosphorylation levels of MARCKS in white blood cells of depression patients.

The parasympathetic nervous system is the physiological opponent of the sympathetic nervous system. A low activity of the parasympathetic nervous system can be quantified by low heart rate variability (HRV). Low HRV has been detected in humans suffering from stress [202] or depression [203,204,205] as well as in suicidal individuals [206]. Depression severity has been observed as being negatively correlated with HRV [203,204]. It would be useful to test if improvements in HRV correlate with amelioration of stress-induced depression symptoms.

Hypercortisolemia and dysregulation of the HPA axis are often found in severe forms of depression [81,207,208,209]. In contrast, atypical depression (characterized by hyperphagia and hypersomnia) is associated with low cortisol levels [207]. Hypocortisolemia is also frequently observed in fibromyalgia, chronic fatigue syndrome, and post-traumatic stress disorder [207]. In patients with depression, high cortisol levels in blood or saliva are associated with future completed suicide [210,211,212], but, remarkably, abnormally low cortisol levels are also a risk factor for suicide [49,198,213]. Moreover, there is evidence that glucocorticoids may diminish suicide numbers [214]. This short overview shows that both high and low cortisol levels are associated with depression and suicide risk, and this of course jeopardizes its utility as a biomarker.

In principle, the same is also true for the biomarker glutamine-synthetase. The enzyme glutamine-synthetase (glutamate ammonium ligase (GLUL)) is exclusively expressed in astrocytes and promotes the synthesis of glutamine from glutamate (reviewed: [215]). Astrocytes play a central role in both stress-induced activation of microglia and inflammation-induced neuron loss (Figure 1 and Figure 2). LPS and inflammatory cytokines inhibit glutamine-synthetase activity and also, as a consequence, glutamate uptake, glutamine synthesis, and neuronal-protection are diminished [216]. Conversely, the neuroprotective effect of astrocytes is increased after forced expression of GLUL [216]. The activity of GLUL in astrocytes is increased by glucocorticoids and glutamate but reduced by glutamine [217]. Dexamethasone has been found to induce an increase in synthesis and activity of GLUL in astrocytes in culture, whereas noradrenaline, in itself ineffective, potentiated GLUL activity [217]. Since GLUL activity is also stimulated by glutamate, the data by Hansson ([217]) indicate that all three stress factors (noradrenaline, corticosterone, and glutamate) increase GLUL function. Thus, ‘stress’ (Figure 1) and ‘inflammation’ (Figure 2) display opposite effects on GLUL function in astrocytes. Lithium, via an increase in β-catenin, activates the transcription of the GLUL gene [218,219]. Notably, β-catenin levels are diminished in post mortem brains of suicide victims [220], whereas Li has an anti-suicidal activity [221,222]. The proposal that low GLUL levels/activity could be a biomarker for suicide [215] is therefore jeopardized, due to the finding that GLUL levels become elevated under stress conditions.

### 5.3. Biomarkers for Inflammation-Induced Effects

The value of cortisol or GLUL levels as biomarkers for depression and suicide could eventually be rescued if we were able to distinguish between stress-induced and inflammation-induced consequences (Figure 3). While stress is thought to lead to depression with melancholic symptoms, inflammation tends to result in atypical depression [147,223]. Roughly one third of MDD patients have clearly elevated cytokines [224,225,226,227]. Patients in this subgroup are frequently obese [225,227,228] and/or suffer from metabolic syndrome [98,229] and cardiovascular disease [96]. They often display atypical features such as increased appetite, hypersomnia, and fatigue [230]. Their depression symptoms may preferentially respond to dietary interventions such as dietary restriction or the fish-oil component eicosapentaenoic acid [227,231]. It may well be that patients with atypical depression are the ones that display a distinct suicide-endophenotype with increased extraversion, impulsivity, and violent attempts [109,115,116]. Infectious agents such as human immunodeficiency virus (HIV), neuro-borreliosis, and *Toxoplasma gondii* are associated with agitation, aggression, and violent suicide attempts [224,226]. Numerous inflammatory mediators [115,228], including IL-6 in its trans-signaling mode, increase the expression of the enzyme indoleamine 2,3-dioxygenase (IDO) [156]. IDO converts L-tryptophan to L-kynurenine, and this is further metabolized to quinolinic acid. In suicidal individuals increased levels of IL-6 [108,114], L-kynurenine [232] and quinolinic acid [233] have all been reported. An IDO-induced increase in the production of kynurenine, and quinolinic acid have negative consequences for the availability of tryptophan for serotonin and melatonin synthesis. Consequently, CSF, plasma, and urine levels of the serotonin metabolite 5-hydroxy-indolic acid (5HIAA) are often strongly diminished in suicidal individuals [234]. The same is true for melatonin [235,236]. The enzymatic activity of IDO can be measured by a relatively simple, non-invasive method. Teraishi et al. ([237]) orally administered C^13^-labeled tryptophan to MDD patients and respective controls. Exhaled ^13^C-CO_2_ was quantified over the next three hours. Compared to the controls, in the MDD patients the recovery rate and peak levels of ^13^C-CO_2_ were significantly larger. This method would also be suitable to study IDO-activity in patients at an increased risk of suicide.

## 6. General Remarks

From the above it is evident that the subgroup of patients with major depressive disorder that suffer from an inflammation-prominent form of the disorder can be easily identified by multiple biomarkers (cytokine levels, IL-6-trans-signaling, low 5HIAA, low GLUL, low(er) cortisol, high kynurenine and quinolinic acid, and high tryptophan metabolism). Putative novel treatments for these particular patients are sGP130fc (particularly in case of an inflammation outside the brain), or TYK2-inhibition. The remaining group of MDD patients is probably divided in a group defined by stress-induced depression, and a group with a mix of inflammation and stress. High cortisol levels, high α-amylase, PKC-activity, MARCKS-phosphorylation, and high GLUL are tentatively biomarkers for this group of patients with stress-induced depression. Novel treatments for this group could comprise low-dose α2-adrenoceptor agonists, β-blockers, and treatments that reduce high intracellular chloride levels (shifting the effect of GABA from depolarizing to inhibitory). Other treatment targets for this group of patients could be NMDA-blockade, P2X4-blockade, inhibition of the BK-potassium channel, and/or GSK3-inhibition. Vagus nerve stimulation and nicotinic α7 receptor agonists are particularly interesting targets because these interventions might improve both stress- and inflammation-induced symptoms and would be particularly suited to patients in whom depression is driven by a mix of stress and inflammation factors. The information collected in the current review could be used to generate and improve sequential treatment optimization paradigms, such as for instance that described by Kraus et al. [238].

## Figures and Tables

**Figure 1 pharmaceuticals-12-00113-f001:**
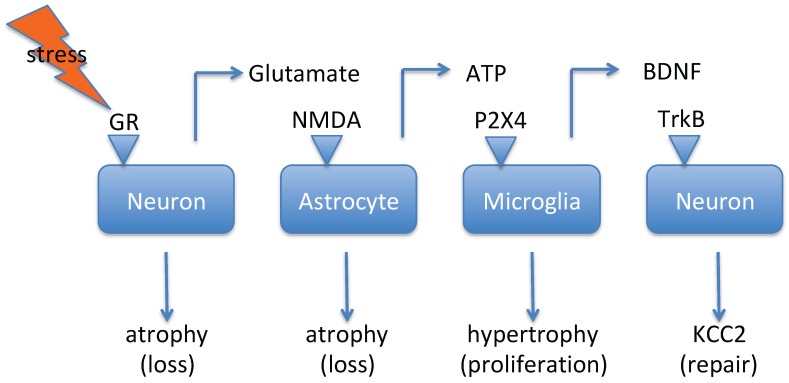
This scheme is based upon data published by [44,58]. Stress activates a sequence of events that involves neuron-derived glutamate, astrocyte-derived ATP, and microglia-derived brain-derived neurotrophic factor (BDNF), and ultimately results in higher network excitability. Since stress induces dendritic atrophy, the elevated network excitability might serve a repair function. Legend: GR, glucocorticoid receptor; NMDA, N-methyl-D-aspartate; TrkB, tropomyosin receptor kinase-B; KCC2, K-Cl-cotransporter-2.

**Figure 2 pharmaceuticals-12-00113-f002:**
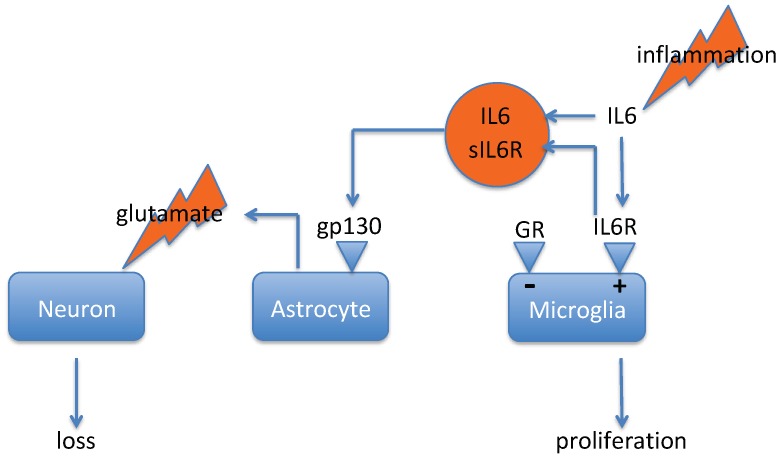
This scheme is based upon data published by [143]. It assumes that inflammation causes interleukin-6 (IL6)-release and shedding of sIL6R from microglia. Trans-activation of astrocytes would lead to the ‘neurotoxic-reactive’ phenotype that may cause damage to neurons. Activation of the glucocorticoid receptor would limit the inflammation-induced sequence of events. Legend: sIL6R, soluble IL-6-receptor.

**Figure 3 pharmaceuticals-12-00113-f003:**
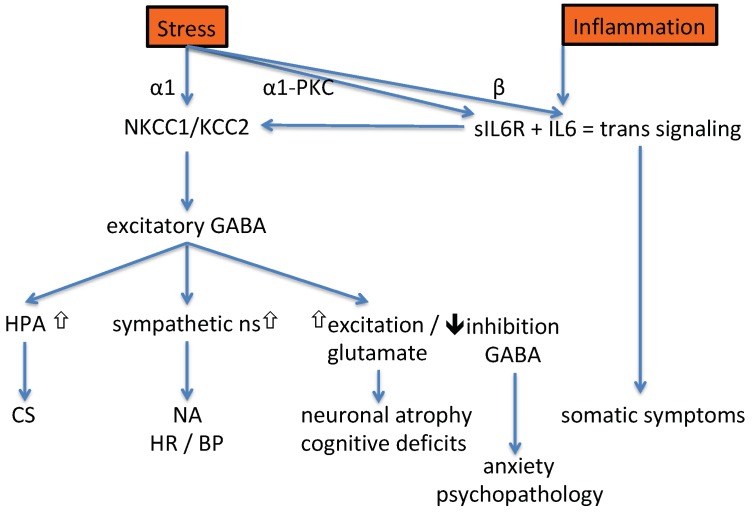
A graphical summary of the biological effects induced by stress and inflammation. As reviewed in Section 2 and Section 3 of this manuscript, inflammation and psychological stress activate the hypothalamic-pituitary-adrenal (HPA) axis, the sympathetic nervous system, and increase glutamate, while the inhibitory effect of gamma-aminobutyric acid (GABA) is diminished. This leads to measurable increases in circulating corticosteroids, catecholamines, anxiety, somatic and psychopathological symptoms, and a decline in cognitive functions. Both stress and inflammation affect the function of the chloride transporters, Na-K-Cl-cotransporter-1 (NKCC1) and K-Cl-cotransporter-2 (KCC2), and provoke IL-6 trans-signaling. Although there is a large qualitative overlap, it may be that stress and inflammation lead to quantitative differences in the severity of the individual effects. Legend: BP = blood pressure, CS = corticosterone, HR = heart rate, NA = noradrenaline, ns = nervous system, PKC = protein kinase C, sIL6R = soluble IL6 receptor.

**Table 1 pharmaceuticals-12-00113-t001:** Depression is comorbid with inflammatory disorders.

Comorbidity	Citation
Viral infection (e.g., HIV*)	[36,81]
Bacterial infection (e.g., periodontitis)	[68,82]
Allergic inflammation (e.g., asthma)	[83]
Autoimmune disease (MS)	[84]
Autoimmune disease (RA)	[85,86,87]
Neurological disorder (Parkinson’s)	[88,89]
Neurological disorder (Alzheimer’s)	[90,91]
Cardiovascular disease (heart failure)	[92]
Diabetes	[93]
Obesity	[94]

*HIV, human immunodeficiency virus; MS, multiple sclerosis; RA, rheumatoid arthritis.

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
