# Peer review of "Novel Treatment Targets Based on Insights in the Etiology of Depression: Role of IL-6 Trans-Signaling and Stress-Induced Elevation of Glutamate and ATP"

_pharmaceuticals, 2019, doi:10.3390/ph12030113_

Round 1

Reviewer 1 Report

This manuscript covers a significant amount of information discussing the importance of stress and inflammation underlying depression. There are a few minor edit suggestions to consider. 

The title is broad for the material/content discussed throughout the manuscript. 
Consider incorporating aspects of neurotransmitter and cytokine (IL6) biology into the title. 

Spelling out all acronyms initially (ex: TrkB line 122). 

Section 2.3 "Stress Reduces Heart Rate Variability" attenuates the flow of the manuscript and the content.  Consider removing the section from the current position.  This will allow a more effective transition between section 2.3 and 2.5  connecting inflammation and cytokine  IL6) regulation with stress. 

Consider shortening section 3 (IL6 signaling), sections 4 and 5. Specifically, sections 4 and 5 would be easier to read with sub-grouping (headings as markers) of content where possible. 

Author Response

I would like to thank the referees for their time and effort to review my manuscript: “Novel treatment targets based on insights in the etiology of depression”.

I have revised the title (added a subtitle) as requested by referee 1. It is now:  Novel treatment targets based on insights in the etiology of depression: Role of IL-6 trans-signaling and stress-induced elevation of glutamate and ATP” 

I have spelled out all acronyms upon their first occurrence (except for the abstract). 

I have also taken up point 3 by referee 1, and removed the section on heart rate variability. Since HRV is a potential biomarker, I have transferred this section, in shortened form to the biomarker paragraph. Moreover, I have added subheadings to the treatment-paragraph and the biomarker paragraph (as suggested). 

Finally I have slightly shortened the IL6-signaling section (e.g. removed the mentioning of meprin as IL6R protease).

I think that with the help of the referees, the current form of the manuscript is definitively better than the previous. And, I am grateful for that.   

Reviewer 2 Report

In this manuscript, Kalkman has reviewed novel treatment targets based on insights from the etiology of depression, as the title suggests. The review is well written, and several aspects of stress and inflammation related depression and its targets are discussed. The author has tried to cover the majority of studies but in doing so, have discussed a few aspects superficially. SERPIN is mentioned in the abstract but not discussed. I would recommend taking a few cases (for ex., IL-6 signaling) and discuss the pathophysiological and social imperative with respect to developing new therapeutics.

1.     Inflammation can also lead to stress and vice-versa is also true. There are several nice studies that have separated inflammation related depression to stress. This will give readers an unbiased opinion on depression.

2.    Heart patients are more likely to suffer from depression, and the opposite is also true. There are emerging studies suggesting a link between these two conditions. Discussing more on the clinical front (heart disease, rheumatoid arthritis, etc.) covering a broad range of disorders connected with depression will definitely give more impact to this review. To simplify this, a table can be added to present all the disorders linked to depression; their inflammatory cues and targeted therapies with original references.

3.    Adding a bit of evolutionary perspective to the role of inflammation in depression and a timeline to the development of pharmaceutics would be interesting to a broad readership.

4.    IL6 should be replaced by IL-6 throughout the manuscript.

(suggested readings: PMID: 30366684; 26711676; 27502736)

Author Response

I would like to thank the referees for their time and effort to review my manuscript: “Novel treatment targets based on insights in the etiology of depression”.

The comments by referee 2 are much appreciated. They clearly reflect the thoughts of a clinician. My intention for the current manuscript was to outline potential novel treatment targets, not so much to give an account on clinical aspects and theories of depression. 

This referee will notice that I have maintained the mentioning of SERPIN. This is because it is a gene that is transcribed following IL6 trans-signaling, but not by IL6 classical signaling, and therefore could be a potential biomarker for trans-signaling. 

The social imperative to develop new therapies is something that I have not emphasized further. Since the current manuscript will be a contribution to a special issue on depression, I think this is something that has to be (and sure will be) covered by the editors of this special edition. 

I have added, as requested, a table that summarizes the inflammatory disorders that are frequently co-morbid with depression. 

The evolutionary perspective on the conservation of pro-inflammation alleles in relation to depression (Miller and Raison, 2016) has been added. However, I have not extended this too much, as it is slightly outside the scope of the current manuscript (and moreover, the way as it is formulated by Miller and Raison is simply superb, and ways better than I could do this). 

I have considered adding a hyphen to IL6, but have decided against that. In case of the formulation “IL6-induced”, I would get then two hyphens. And, when IL6 is part of a signalling-sequence, the too many hyphens become confusing. On its first occurrence, I have defined that interleukin-6 will be further abbreviated as “IL6”. 

Finally, I have read the “suggested readings”, and integrated them into the new manuscript. 

I think that with the help of the referees, the current form of the manuscript is definitively better than the previous. And, I am grateful for that.   

Round 2

Reviewer 1 Report

The updated version of the manuscript, with the minor edits, provides a greater sense of clarity for the scientific content presented. 

It is a comprehensively put together piece of contribution put to the field. 

Reviewer 2 Report

The revised manuscript has been improved.